# Curcumin-Loaded RH60/F127 Mixed Micelles: Characterization, Biopharmaceutical Characters and Anti-Inflammatory Modulation of Airway Inflammation

**DOI:** 10.3390/pharmaceutics15122710

**Published:** 2023-11-30

**Authors:** Xinli Wang, Yanyan Wang, Tao Tang, Guowei Zhao, Wei Dong, Qiuxiang Li, Xinli Liang

**Affiliations:** 1Key Laboratory of Modern Preparation of TCM, Ministry of Education, Jiangxi University of Chinese Medicine, Nanchang 330004, China; sofia13ts@gmail.com (X.W.); weiweihaoyunqi@163.com (G.Z.); sober96@foxmail.com (W.D.); liqiuxiang76@163.com (Q.L.); 2Jiangxi Medical Device Testing Center, Nanchang 330029, China; 3Clinical Medical School, Jiangxi University of Chinese Medicine, Nanchang 330004, China; guyuan97@163.com; 4Department of Pharmacy, Ji’an Central People’s Hospital, Ji’an 343000, China; marryrose1013@163.com

**Keywords:** curcumin, nanomicelles, bioavailability, airway inflammation

## Abstract

Curcumin’s ability to impact chronic inflammatory conditions, such as metabolic syndrome and arthritis, has been widely researched; however, its poor bioavailability limits its clinical application. The present study is focused on the development of curcumin-loaded polymeric nanomicelles as a drug delivery system with anti-inflammatory effects. Curcumin was loaded in PEG-60 hydrogenated castor oil and puronic F127 mixed nanomicelles (Cur-RH60/F127-MMs). Cur-RH60/F127-MMs was prepared using the thin film dispersion method. The morphology and releasing characteristics of nanomicelles were evaluated. The uptake and permeability of Cur-RH60/F127-MMs were investigated using RAW264.7 and Caco-2 cells, and their bioavailability and in vivo/vitro anti-inflammatory activity were also evaluated. The results showed that Cur-RH60/F127-MMs have regular sphericity, possess an average diameter smaller than 20 nm, and high encapsulation efficiency for curcumin (89.43%). Cur-RH60/F127-MMs significantly increased the cumulative release of curcumin in vitro and uptake by cells (*p* < 0.01). The oral bioavailability of Cur-RH60/F127-MMs was much higher than that of curcumin-active pharmaceutical ingredients (Cur-API) (about 9.24-fold). The treatment of cell lines with Cur-RH60/F127-MMs exerted a significantly stronger anti-inflammatory effect compared to Cur-API. In addition, Cur-RH60/F127-MMs significantly reduced OVA-induced airway hyperresponsiveness and inflammation in an in vivo experimental asthma model. In conclusion, this study reveals the possibility of formulating a new drug delivery system for curcumin, in particular nanosized micellar aqueous dispersion, which could be considered a perspective platform for the application of curcumin in inflammatory diseases of the airways.

## 1. Introduction

Cremophor^®^ RH60 (RH60) is a polyethoxylated hydrogenated castor oil that contains 60 ethoxy units in its polyethoxylated chain. It is an amphiphilic polymeric surfactant composed of glycerol and polyethylene glycol hydroxystearate, with a lipophilic end of fatty acid glycerol polyethylene glycol ester and a hydrophilic end of polyethylene glycol and glycerol ethoxylate. They can spontaneously form nanomicelles in water and have a high solubilization capacity, high transmembrane transport efficiency, and good biocompatibility [1,2]. These nanomicelle polymers self-assemble to form nanoparticles with a core–shell structure. The hydrophobic chain segments form nuclei on the inside, which serve as repositories for insoluble pharmaceuticals, while the hydrophilic chain segments form shells on the surface, which provide stability to the particles [3]. Pluronic F127 is a triblock copolymer with a central hydrophobic poly- (propylene oxide) (PPO) chain with hydrophilic poly- (ethylene oxide) (PEO) chains attached on each side, whose PPO chain forms a hydrophobic nucleus that effectively increases drug solubility, and the PEO chain forms a hydrophilic shell with a stable spatial effect that prolongs drug circulation in vivo [4]. The drug loading capacity and stability of nanomicelles were found to be important formulation characterization parameters. Furthermore, particle size distribution and charge are important parameters for evaluating micelles, which are closely related to the ADME (absorption, distribution, metabolism, and excretion) of micelles in vivo [5,6]. However, some single polymer micelles suffer from performance problems such as limited drug loading capacity and insufficient in vivo stability due to the structural type and length limitations of polymer molecules. Therefore, using a combination of two or more amphiphilic polymers to construct mixed polymer micelles (MMs) is an effective method for addressing these issues [7].

Curcumin (Cur) is a diketone-active chemical compound that is mainly extracted and isolated from the traditional herbal medicine *Turmeric rhizome*. Modern pharmaceutical studies have identified that Cur has good biological activity, including hypolipidemic, choleretic, anti-inflammatory, antioxidant, and antitumor effects [8,9,10,11,12,13,14,15,16]. In addition, Cur has low toxicity and is also safe at 8.0 g/kg according to clinical trials [17]. On the other hand, Cur is a BCS class IV medication that is insoluble in water, has poor membrane permeability and stability, and undergoes significant metabolic transformations in the intestine and liver, resulting in essentially negligible oral absorption, and limiting its clinical application. Cur has a maximum solubility of 11 ng/mL in aqueous solution at pH 5.0, with an oral bioavailability of barely 1% in rats [18], and clinical trials require a dosage of 3.6 g to produce measurable serum levels [18]. Therefore, improving the solubility and permeability of Cur, as well as its oral bioavailability, has always been a difficulty and challenge for researchers.

Based on this, the micelle particle size, potential, and drug loading were taken as evaluation indexes to investigate the single and mixed micelles of RH60 and F127 using a film dispersion method. The results showed that the mixed micelles had a more uniform particle size distribution and higher drug loading. Then, Cur-RH60/F127-MMs were prepared to optimize the RH60 and F127 ratios in the previous phase of this project. In this study, Cur-RH60/F127-MMs were physicochemically characterized, and drug release was investigated using the dialysis method in vitro. The uptake of Cur-RH60/F127-MMs by two cell types (Caco-2 and RAW264.7) and the absorption of Cur was investigated using the Caco-2 cell model in vivo. The oral bioavailability of Cur-RH60/F127-MMs in rats was evaluated. In this paper, the inflammatory model of RAW264.7 macrophage induced by LPS and INF-γ and the asthma model of BALB/c mice induced by OVA were used to study the anti-inflammatory effect of Cur-RH60/F127-MMs in vitro and in vivo and its therapeutic effect on bronchial asthma in mice.

## 2. Materials and Methods

### 2.1. Chemicals and Reagents

Curcumin (Batch No. AF9102903, Purity ≥ 98%, Alfa Biotechnology Co., Ltd., Chengdu, China); cremophor^®^ RH60 (BASF, Ludwigshafen, Germany); pluronic F127 (CRODA, Shanghai, China; Keke tablets (Heng Sheng Pharmaceutical Co., Ltd., Zhongshan, China).

DMEM high-sugar medium (batch NO. 12100, Solarbio, Beijing, China); DMEM complete medium (batch NO. ZQ-120, Zhongqiao Xinzhou Biotechnology Co., Ltd., Shanghai, China); Fetal bovine serum (FBS, Qualified, Lot No. 2152050CP, Gibco, Thermo Fisher Scientific, Rockford, IL, USA); IFN-γ (lot NO. Z02916-20, Genscript, Piscataway, NJ, USA); Ovalbumin (OVA, lot NO. MO228A, Meilun Biotechnology Co., Ltd., Suzhou, China); Aluminum hydroxide gel (lot NO. WB317588, Thermo Fisher Scientific, Rockford, IL, USA); PBS (Lot No. P1020), D-Hank’s buffer (Lot No. H 1045), Lipopolysaccharide (LPS, lot NO. 426Y031), Dimethyl sulfoxide (DMSO, Dimethyl sulfoxide, lot NO. 710NO034), Thiazole blue (MTT, lot NO. 1223G0531) were purchased from Solarbio Co., Ltd. (Beijing, China). Nitric Oxide Assay Kit (Batch No. S0021S), Lipid Oxidation (MDA) Assay Kit (Batch No. 080521211115), and Total SOD Activity Assay Kit (Batch No. 071521211011) were purchased from Beyotime Biotechnology Co., Ltd. (Shanghai, China). TNF-α ELISA Kit (Batch No. 22A250), IL-6 ELISA kits (lot NO. 22A256), and IL-10 ELISA kits (lot NO. 22A267) were purchased from Excell Co., Ltd. (Shanghai, China). A Mouse immunoglobulin E kit (IgE, lot NO. 20210922, Jiancheng Institute of Biological Engineering, Nanjing, China) and Mouse intercellular adhesion molecule-1 (ICAM-1) ELISA kit (lot NO. 202112, Meimian Industrial Co., Ltd., Yancheng, China) were also acquired.

### 2.2. Animals and Cell Culture

Twelve male Sprague Dawley (SD) rats (180–220 g) were purchased from Slack King Experimental Animal Center in Hunan (Changsha, China). Fifty SPF-grade female BALB/c mice (6 weeks, 18–22 g), were purchased from Spelford Biotechnology Co., Ltd. (Beijing, China). Before the experiment, all animals were housed for 1 week in an environmentally controlled room 25 ± 2 °C and 52 ± 20% RH with free access to food and water. 

RAW264.7 cells were purchased from the Cell Bank of the Chinese Academy of Sciences, Caco-2 cells were purchased from Shanghai FuHeng Biotechnology Co., Ltd. (Shanghai, China). All cells were maintained at 37 °C in an atmosphere containing 5% CO_2_.

### 2.3. Preparation of Cur-RH60/F127-MMs

Cur-RH60/F127-MMs were prepared using the film hydration method. Cremophor^®^ RH60 (1.6 g), F127 (0.4 g), and Cur (500 mg) were dissolved in anhydrous ethanol (10 mL) and vortexed for 10 min to mix thoroughly. Then, the anhydrous ethanol was removed by rotary evaporation at 50 °C and vacuum-dried overnight to obtain a light-yellow semisolid film. Distilled water (25 mL) was added to the hydrated resulting film for 30 min at 37 °C, and sonicated (ultrasonic power: 200 W; ultrasonic temperature: 37 °C) for 10 min. After leaving for 1 h at room temperature, the Cur-RH60/F127-MMs were filtered through a 0.22 μm microporous filter membrane. Blank MMs (without drugs) were prepared as described above.

### 2.4. Particle and Zeta Potential Analysis, Tyndall Effect, and Micromorphological Observation of Cur-RH60/F127-MMs

The particle size, polydispersity index, and Zeta potential of the Cur-RH60/F127-MMs were measured using a Malvern laser particle size analyzer (Malvern, Worcestershire, UK), after the samples (1 mL) were adequately diluted with distilled water.

Samples (5 mL) were pipetted into a cillin vial, and a laser pointer from one side was used to investigate the Tyndall effect of Cur-RH60/F127-MMs.

The morphologies of the Cur-RH60/F127-MMs were observed using a transmission electron microscope (TEM, JEOL, Tokyo, Japan). After dilution with distilled water, a drop of solution was placed on the carbon-sprayed copper grid for 10 min, and negatively stained with 1 percent phosphotungstic acid for 2 min.

### 2.5. Determination of Encapsulation Yield (EY) and Drug Loading (DL) Capacity

EY was determined using ultra-high-speed centrifugation. In addition, the DL and EY were determined following the solubilization of carriers in ethanol and analysis using the high-performance liquid chromatography (HPLC) method. The chromatographic conditions were as follows: Phenomenex-C_18_ (4.6 mm × 250 mm, 5 μm); the mobile phase consisted of acetonitrile 0.5 percent acetic acid water (58:42, *v*/*v*). A volume of 10 µL of the sample was injected, and the flow rate was 1 mL·min^−1^. The column temperature was maintained at 25 °C, and the detection wavelength was set at 430 nm.
DL = [W_m_/(W_c_ + W_f_)] × 100%
EY(%) = (W_m_/W_f_) × 100%
where W_m_ stands for the drug dosage in micelles, W_c_ for carrier dosage, and W_f_ for drug delivery dosage.

### 2.6. Determination of Blank MMs Critical Micelle Concentration (CMC)

The pyrene fluorescence probe method was used to evaluate the critical micelle concentration (CMC) values of RH60, F127, and RH60/F127 blank MMs, respectively. A total of 1 mL of pyrene solution (0.012 mg·L^−1^) was, respectively, added into 10 volumetric flasks (5 mL), and blow-dried with nitrogen at 40 °C. Then, the prepared micellar solution was added to them according to a certain volume, and the volume was fixed to the scale line with ultrapure water to prepare a series of different concentrations of pyrene micellar solutions. Finally, all blank MMs were sonicated for 40 min at 65 °C in a constant temperature water bath and left for 8 h at room temperature, which was protected from light.

The fluorescence emission spectrum of pyrene solution has five distinct peaks from 373, 379, 384, 394, and 480 nm. The CMC value of the micelles was plotted using the ratio of pyrene fluorescence intensity at 373 nm in the first peak to 384 nm (I_373nm_/I_384nm_) in the third peak as the *Y*-axis and the logarithm of micelle concentration (log C) as the *X*-axis, and the mass concentration corresponding to the intersection point was the CMC value of the micelles.

### 2.7. In Vitro Release

In vitro release behaviors of curcumin API (Cur-API) and Cur-RH60/F127-MMs were monitored using the dialysis bag method with minor modifications. A total of 1 mL of the Cur-API and Cur-RH60/F127-MMs solutions was, respectively, sealed into a dialysis bag (MWCO 12 kDa), and dialyzed extensively under mechanical shaking at 100 r·min^−1^ and (37 ± 0.5) °C. At prefixed time intervals, 1 mL of the samples was withdrawn and replenished with a fresh blank medium (100 mL of PBS with 1% TW-80) [19,20,21]. We separated the released drugs from Cur-API and Cur-RH60/F127-MMs by centrifugation and evaluated them using the HPLC. The cumulative release rate was calculated, the cumulative release curve with time was drawn, and then the origin 8.0 software was used to fit the release curve.

### 2.8. Intracellular Uptake and Trans-Cellular Membrane Transport

RAW264.7 and Caco-2 intracellular uptake: RAW264.7 and Caco-2 cells were cultured on 6-well plates at a density of 3.0 × 10^6^ and supernatant was discarded after 24 h of growth. Different concentrations of Cur-API and Cur-RH60/F127-MMs were added to the cells and cells were subsequently cultured for 4 h. The culture liquid was discarded, and the cells were washed with PBS 3 times. After that, the cells were lysed using RIPA, collected, and measured using HPLC-MS/MS [22,23].

Trans-cellular membrane transport in Caco-2 cells: Caco-2 cell monolayers were cultured on a transwell of 24 wells for 15–17 d and examined by measuring the transepithelial electrical resistance (TEER) with an epithelial volt-ohm meter. Caco-2 cell monolayers were used for the transport experiment only when TEER values reached 350 Ω·cm^2^. The transport experiments were conducted by adding 40 μM of Cur-API and Cur-RH60/F127-MMs solutions to the apical (AP, 0.4 mL) and blank Hank’s to the basolateral side (BL, 0.6 mL). At prefixed time intervals, 100 µL of the samples were withdrawn and replenished with fresh blank Hank’s [20,23]. The samples were evaluated using HPLC-MS/MS, and the apparent permeability coefficient P_app_ was calculated according to the following formula:$$P_{app}=\frac{dQ/dt}{{AC}_{0}}\times 100\%$$
where *dQ*/*dt* (μg/s) is the slope of the linear regression function of the cumulative number of transfers versus time, *A* is the surface area of the transfer chamber (0.33 cm^2^), and *C*_0_ is the initial concentration of the drug delivery chamber.

### 2.9. Pharmacokinetic Study

The SD rats were randomly divided into two groups of six rats, each given Cur-API suspension (dissolved in 0.5% CMC-Na) and Cur-RH60/F127-MMs administered intragastrically at a dose of 60 mg/Kg. Orbital blood samples (0.5 mL) were collected at 0.083, 0.25, 0.5, 0.75, 1, 1.5, 2, 3, 4, 6, 8, and 12 h after administration. Blood samples were placed in heparinized tubes and immediately centrifuged in a centrifuge tube coated with sodium heparin at 10,000 rpm at 4 °C for 10 min. The supernatant was taken and stored at −80 °C until further analysis.

Plasma samples were treated with the liquid–liquid extraction method. A total of 60 μL of plasma samples, 10 μL of the tinidazole control solution (internal standard, 32 ng/mL), and 180 μL of methanol were placed in an Eppendorf tube. Samples were then vortex-mixed for 3 min and centrifuged for 20 min at 15,000 rpm. Approximately, 100 μL of the supernatant was placed into another clean tube and filtered with a 0.22 μm filter. Samples were analyzed using LC-MS/MS [24,25].

### 2.10. In Vitro Antioxidant Research

A series of concentrated solutions of Cur API and Cur-RH60/F127-MMs, and 100 mM of DPPH solution (dissolved in anhydrous ethanol) were prepared, which were stored at 4 °C and kept out of the light. The absorbance at 517 nm of the following samples was measured separately: (i) 100 µL of the sample mixed with 100 µL of the DPPH solution (A_i_); (ii) 100 µL of the sample mixed with 100 µL of the anhydrous ethanol solution (A_j_); (iii) 100 µL of DPPH mixed with 100 µL of anhydrous ethanol solution (A_0_). Additionally, DPPH clearance was calculated according to the following formula [26]:$$DPPH\;clearance\;\left(\%\right)=(1-\frac{A_{i}-A_{j}}{A_{0}})\times 100\%$$

### 2.11. Effect of Cur-RH60/F127-MMs on RAW264.7 Cells Stimulated by LPS

RAW264.7 cells were cultured in 24-well plates at a density of 2 × 10^5^, and the supernatant was discarded after 24 h of growth. The study itself followed a simple parallel design in which cells were allocated to five groups: normal control group (NC), model control group (MC), positive control group (PC, 2 μM of dexamethasone, DEX), and two treatment groups (5 μM of Cur-API and Cur-RH60/F127-MMs). The positive control group and treatment groups were, respectively, pretreated for 1 h, and a medium without drugs was added to the normal control group. Except for the blank group, all groups were given LPS (1 g/mL) and INF-γ (20 ng/mL) to induce cellular inflammation, and cell supernatants were collected after 24 h of incubation [27,28,29,30,31,32,33]. The Griess method was used to detect the level of NO; ELISA (Enzyme-linked Immunosorbent Assay) was used to detect the levels of the produced IL-6, IL-10, TNF-α, and IL-1β, according to the manufacturer’s instructions.

### 2.12. Effects of Cur-RH60/F127-MMs on Airway Inflammation and Cytokines in OVA-Induced BALB/c Mice Asthma Model

The BALB/c mice were randomly allocated into five groups: the normal control group, model control group, positive control group (1.4 g/kg of Keke Tablets), and drug-treated groups (20 mg/kg of Cur and Cur-RH60/F127-MMs).

Except for the blank group, 200 µL of 0.9% physiological saline-prepared OVA (50 µg) and aluminum hydroxide (2 mg) solutions were sensitized by intraperitoneal injection on days 1, 7, and 14. On days 16, 17, 19, 21, and 23, 5 mL of OVA solution (50 mg/mL, dissolved in 0.9% physiological saline) was given for 30 min to promote asthma development [30,34,35,36] (Figure 1). The treatment groups were administered intragastrically 30 min before the first stimulation of asthma and then were given it once a day for the rest of the study. The model control group was given the same amount of saline as the blank group.

### 2.13. Cell Count and IgE Level

The mice were killed twenty-four hours after the last treatment. Orbital blood samples (100 µL) were collected in EDTA anticoagulation tubes, while the rest were placed in ordinary Eppendorf tubes for 1 h at room temperature to separate the serum. A whole-blood application analyzer was used to count the number of leucocytes, monocytes, lymphocytes, and granulocytes. Sera were prepared using Giemsa stain, and the number of eosinophils was observed under an inverted microscope. The IgE level of asthmatic mice was detected using ELISA.

### 2.14. Histopathology of the Lungs and Cytokine Levels in Bronchoalveolar Lavage Fluid (BALF)

The thoracic cavity of each group of mice was opened by blunt dissection 24 h after the last nebulization, where the lungs were exposed and the left hilum was ligated. Then, the lobe was taken and HE staining was performed, and the infiltration of lung inflammatory cells was observed under a microscope to assess the level of tissue inflammation.

The trachea was split, then cut into a “v” form, and the right major bronchus was ligated using a surgical suture that ran through the opening between the mouse’s trachea and esophagus. The mouse’s right lung was injected with 0.3 mL of 0.9% NaCl solution, and the BALF was collected after three rounds of gentle pressure and slow withdrawal. The BALF was centrifuged at 1000× *g* at 4 °C for 10 min, and the supernatant was used to measure the levels of IL-6 and TNF- using the ELISA kit.

### 2.15. Determination of SOD Activity, MDA Concentration, and Intercellular Adhesion Factor ICMA-1 in Lung Tissues

The mouse right lung tissues were cut into pieces, then added to PBS at a ratio of 1:9, homogenized in an ice bath, centrifuged at 4 °C, 12,000× *g* for 10 min, and the supernatant was extracted. The concentration of malondialdehyde (MDA) and enzyme activity of superoxide dismutase (SOD) in the samples were determined according to the instructions of the lipid peroxidation MDA and SOD assay kit.

Another right lung tissue homogenates were centrifuged at 4 °C, 3000 rpm for 20 min, the supernatant was removed, and the precipitate was washed 3 times with PBS. The tissue cells were suspended and adjusted to a density of 10^5^ cells·mL^−1^ in PBS.

The cells were crushed with ultrasonic waves in an ice bath, centrifuged at 4 °C, 3000 rpm for 20 min, and the supernatant was extracted. The concentration of ICAM-1 was calculated according to the instructions of the ICAM-1 ELISA assay kit.

### 2.16. Data Analysis

All experimental data in this experiment were expressed as the mean ± standard error. An independent samples *t*-test and one-way ANOVA were used to analyze the significance of the difference between the groups using SPSS Statistics software (v24.0, SPSS Inc., Chicago, IL, USA). Data analysis of pharmacokinetic parameters was performed using the DAS 3.3.0 pharmacokinetic program (Chinese Pharmacology Society, Shanghai, China). 

## 3. Results and Discussions

### 3.1. Characterization of Cur-RH60/F127-MMs

As shown in Figure 2a,b, the particle size of Cur-RH60/F127-MMs was 11.23 ± 0.62 nm, PDI was 0.153 ± 0.007, and Zeta potential was −7.62 ± 0.13 mV, all of which have a good normal distribution. Figure 2c depicts the Tyndall effect. Cur-RH60/F127-MMs was spherical, regular in shape and narrow and uniform in distribution under TEM (Figure 2d, which is consistent with the particle size results.

The DL was calculated according to the standard curve (A = 42.701C–38.642, R^2^ = 0.9999). The EY of Cur-RH60/F127-MMs was 89.43 ± 3.07%, while drug loading was 19.78 ± 0.31% (*n* = 3). Because Cur is encapsulated as hydrophobic tiny molecules in self-assembled micelles and has very poor solubility, extremely small amounts of free Cur were not evaluated. 

### 3.2. Determination of Blank MMs CMC

I_373nm_/I_384nm_ is frequently used to characterize the polarity of the environment in which the pyrene probe is located, with a larger value indicating a more polar microenvironment for the flower probe and a smaller I_373nm_/I_384nm_ indicating a less polar microenvironment. Figure 3 shows CMCs of 0.82, 0.002, and 0.004 mg/mL for F127, RH60, and RH60/F127-MMs, respectively, which are consistent with the literature [37,38]. Due to the large amount of RH60 in the Cur-RH60/F127-MMs (80%), the CMC of RH60/F127-MMs was biased toward RH60.

### 3.3. In Vitro Curcumin Release Profile

The release profiles of the Cur-API and Cur-RH60-MMs curves were plotted according to the cumulative release percentage in Figure 4. The release profile of Cur-RH60-MMs was lower than that of Cur-API during the first 6 h. Followed by a sustained release pattern during the 96 h testing period, the release profile of Cur-RH60-MMs was significantly higher than that of Cur-API. Cur-RH60-MMs remained largely intact for 6 h and could effectively avoid drug degradation by gastrointestinal fluid and facilitate micelle absorption through the membrane in their intact form, enhancing the slow-release effect of the drug. With the increase in release time, the cumulative release of curcumin from Cur-RH60-MMs was significantly higher than that of Cur-API. According to the release curve equation, fitting was performed on the in vitro release data of two groups of preparations; in the fitting equation, the closer the correlation coefficient is to 1 (that is, R^2^ ≈ 1), the more appropriate the model is. The result showed (Table 1) that the releasing behavior of Cur-API followed the Ritger–Peppas equation (F = 2.1951t^0.3881^, Rsqr_adj = 0.9424), while the releasing behavior of Cur-RH60/F127-MMs confirmed the Higuchi equation (F = 5.9356t^1/2^ − 4.936, Rsqr_adj = 0.9506). The results showed that there is little difference between the release of Cur-API and Cur-RH60/F127-MMs, and Cur-RH60/F127-MMs have the characteristics of slow diffusion and release in the medium.

### 3.4. Intracellular Uptake and Trans-Cellular Membrane Transport Study

An important aspect of the evaluation of a potential drug delivery system is intracellular uptake and transcellular membrane transport. These will shed light on whether the drug delivery system is effective in vivo. As presented in Figure 5a,b, Cur-RH60/F127-MMs were more effective in the enhancement of Cur-API uptake in RAW264.7 cells and Caco-2 cells (*p <* 0.001). Additionally, Cur-RH60/F127-MMs significantly increased the permeability of Cur into the cell membrane, as shown in Figure 5c (*p <* 0.01).

### 3.5. Pharmacokinetic Study

The mean plasma concentration–time curves are shown in Figure 6. The major pharmacokinetic parameters are listed in Table 2.

When compared with Cur-API, the AUC_(0–12)_ of Cur-RH60/F127-MMs significantly increased (9.24 times of Cur-API, *p* < 0.05), indicating that Cur-RH60/F127-MMs exceedingly enhanced the oral absorption of Cur. The peak (maximum) plasma concentration (C_max_) of Cur-RH60/F127-MMs (199.84 ± 39.05 ng·mL^−1^) was 10.80 times that of Cur-API (18.50 ± 1.81 ng·mL^−1^), while Cur-RH60/F127-MMs showed a shorter half-life (t_1/2_ = 1.98 ± 0.91) and clearance of (181.97 ± 54.58) L·h^−1^·kg^−1^ than that of Cur-API (t_1/2_ = 5.16 ± 2.76 h, CLz/F = 982.85 ± 314.28 L·h^−1^·kg^−1^). The short half-life suggests that Cur-RH60/F127-MMs should be quickly metabolized in vivo and they should have a short duration of efficacy. The result suggested that we should investigate the prolongation of the half-life of Cur-RH60/F127-MMs.

### 3.6. In Vitro Antioxidant Research

The 1,1-diphenyl-2-picrylhydrazyl (DPPH) radical scavenging activity was used to assess the antioxidant potential of Cur API and Cur-RH60/F127-MMs. Cur-API and Cur-Cr RH60/F127-MMs both showed certain DPPH radical scavenging activity, and Cur-Cr RH60/F127-MMs had a positive correlation at 5–100 µM of Cur concentration (Figure 7). The IC_50_ of Cur-RH60/F127-MMs before UV irradiation was 25.80 µM, and after UV irradiation, it was 29.42 µM. The DPPH scavenging ratios of 100 µM Cur-RH60/F127-MMs and Cur-API were 86.55% and 21.09%; after UV irradiation, it was 86.90% and 18.95%. The decrease in DPPH radical ability of Cur-API is mainly due to the decomposition of curcumin by UV light. However, this hardly affects Cur-RH60/F127-MMs, indicating that it improves the ultraviolet decomposition characteristics of curcumin.

### 3.7. Effect of Cur-RH60/F127-MMs on RAW264.7 Cells Stimulated by LPS

As presented in Figure 8, the results show that both Cur-API and Cur-RH60/F127-MMs were found to suppress TNF-α, IL-10, IL-6, and NO expression in inflammatory cell models, while Cur-RH60/F127-MMs significantly reduced IL-10, IL-6, and NO expression when compared to the same dose of Cur-API.

### 3.8. Effects of Cur-RH60/F127-MMs on Airway Inflammation and Cytokines in OVA-Induced BALB/c Mice Asthma Model

Inflammatory cells, such as lymphocytes, monocytes, and granulocytes (eosinophils and basophils), are involved in the inflammatory response. As a result, the counting of inflammatory cells in whole blood can be used to determine inflammation. The counting of monocytes, lymphocytes, and granulocytes significantly increased in the whole blood in the asthmatic mice model (*p <* 0.01, Table 3). The result shows that Cur-API and Cur-RH60/F127-MMs both reduced monocytes, lymphocytes, and granulocytes, but Cur-RH60/F127-MMs had a larger effect than Cur-API. The eosinophil count was observed under an inverted microscope. There were only red blood cells in the normal control group (Figure 9a (a-NC)). After given OVA, there are several neutrophils and some lymphocytes indicating that airway inflammation has occurred (Figure 9a (a-MC)). Compared with the model control group, the neutrophils and lymphocytes in the drug treatment group were reduced (Figure 9a (a-PC, a-Cur-API, a-Cur-RH60/F127-MMs)). What is more, the results indicated that the effect of Cur-RH60/F127-MMs on reducing the expression of eosinophils was very pronounced.

The IgE level was considerably higher in the asthma model mouse group (*p <* 0.01). There was no statistical difference in IgE levels in the Cur-API group compared to the model control group, and Cur-RH60/F127-MMs dramatically reduced serum IgE levels (Figure 9b).

As shown in Figure 10f, the levels of IL-6 and TNF-α were significantly increased in BALF of the asthma inflammation mouse model (*p <* 0.01), and Cur-API and Cur-RH60/F127-MMs both significantly reduced the levels of IL-6 and TNF-α in BALF when compared to the model control groups (*p <* 0.05). Meanwhile, Cur-RH60/F127-MMs significantly reduced the level of TNF-α when compared to Cur-API (*p <* 0.05).

### 3.9. Effects of Cur-RH60/F127-MMs on Pathological Changes of Lung Tissue of OVA-Induced BALB/c Mice Bronchial Asthma Model

The results are shown in Figure 10a–e. The alveolar structure of the normal control group was clear, without obvious inflammatory cell infiltration (Figure 10a), while the OVA-induced asthma model exhibited an abnormal lung tissue structure, alveolar atrophy, and significantly increased inflammatory infiltration, epithelial thickening and inflammatory cell infiltration (Figure 10b). After drug treatment, the pathological changes related to OVA-induced asthma were effectively alleviated (Figure 10c–e). Inflammatory cell infiltration in the OVA-induced asthma model was significantly reduced after Cur-API and Cur-RH60/F127-MMs treatment, but a small amount of inflammation still existed after positive drug treatment (Figure 10c–e).

### 3.10. Effects of Cur-RH60/F127-MMs on ICAM-1, MDA, and SOD Levels in Lung Tissue of OVA-Induced BALB/c Mice Bronchial Asthma Model

Figure 11a shows that the expression of MDA was markedly elevated (*p <* 0.01), while the expression of SOD decreased (*p <* 0.05) in the lung tissue of the asthma model control group compared to the control group. However, the expression of SOD increased in the Cur-API group (*p <* 0.05), and the expression of MDA was cut off in the Cur-RH60/F127-MMs group (*p <* 0.05) compared to the model control group. The level of MDA was reduced in the lung tissue of the Cur-RH60/F127-MMs group compared to the Cur-API group (*p <* 0.05).

The level of ICAM-1 significantly increased in the lung tissue of asthma model mice when compared to the control group (*p <* 0.01), whereas the level of ICAM-1 fell off in Cur-API and Cur-RH60/F127-MMs groups compared to the model control group (*p <* 0.05). Additionally, there was no significant difference between Cur-API and the Cur-RH60/F127-MMs groups (Figure 11b).

## 4. Discussion

F127 is a triblock copolymer with a central hydrophobic poly- (propylene oxide) (PPO) chain with hydrophilic poly- (ethylene oxide) (PEO) chains attached on each side. The PPO chain forms a hydrophobic nucleus that effectively increases drug solubility, and the PEO chain forms a hydrophilic shell with a stable spatial effect that prolongs drug circulation in vivo [4]. The RH60 is a polyethoxylated hydrogenated castor oil that contains 60 ethoxy units in the polyethoxylated chain. Thus, F127 and RH60 both have a PEO chain that forms hydrophilic shells. RH60 has a lower CMC, and the glycerol polyethylene glycol oxystearate, which forms a hydrophobic component with the fatty acid glycerol polyethylene glycol ester, may increase the volume of the micelle’s hydrophobic nucleus, allowing more insoluble drugs to enter the micelle’s hydrophobic nucleus and increasing the EY and DL of MMs.

The mean DL of Cur-RH60/F127-MMs prepared by the film dispersion method was 19.78 %, which was significantly higher than the mean DL of the MMs of Cur (4.70 %) reported in the literature [39]. Additionally, their mean solubility was 18.47 mg/mL, which was significantly higher than the solubility of Cur-API in water of 11 ng/mL [40]. The drug release from MMs is based on diffusion and solubilization, and it is generally sluggish, with the release behavior regulated by the composition of MMs [41]. The release effect of Cur-RH60/F127-MMs is also confirmed in vitro, which is most likely due to the formation of hydrogen bonds between the polyhydroxy groups of Cur and the amino (-NH2) and carbonyl groups on the carrier material’s structure, resulting in a slower diffusion of the drug from the hydrophobic core of the micelle to the release medium.

The particle size of micelles is determined primarily by the nature of the polymer chain and DL, and the ideal micelle can avoid reticuloendothelial system phagocytosis due to oversized particle sizes; thus, particle regulation has a significant impact on cellular uptake, in vivo absorption, and bioavailability [42]. Cur-RH60/F127-MMs have a particle size of 10–30 nm, and thus, they have a higher tissue permeability [43], which can facilitate cellular uptake and transport [42]. However, too small particles are quickly eliminated by glomerular filtration before they reach the cells. This explains why Cur-RH60/F127-MMs have increased cellular uptake and transport, as well as enhanced oral bioavailability and faster elimination in vivo.

Asthma is a chronic inflammatory disease of the airways involving multiple cells and cellular components [44], which is triggered by allergens or other environmental factors and results in the release of a large number of inflammatory cells such as eosinophils, T lymphocytes, macrophages, and neutrophils [35,45,46,47]. Therefore, cytokines have been found to play an important role in the development of allergic airway inflammation [48], and Cur may have a significant effect on allergic airway inflammation by reducing inflammatory cytokine levels and exerting anti-asthma therapeutic effects [49]. Macrophages are the most abundant immune cells in the lungs during the onset of asthma. When macrophages are triggered by inflammation, they secrete cytokines such as TNF-α, IL-1, and IL-6 [50,51,52], and release NO [53]. Therefore, to further confirm the efficacy of Cur-RH60/F127-MMs, the inflammatory cell model was established using LPS combined with IFN-γ to stimulate RAW264.7 and to investigate the effects of Cur-RH60/F127-MMs on cytokines and inflammatory mediators such as NO, as well as to assess their anti-inflammatory effects. The results confirmed that it could reduce the levels of inflammatory cytokines TNF-α, IL-6, and IL-10, and the production of the inflammatory mediator NO; therefore, the effect of the formulation was significantly better than that of the Cur-API.

Asthma pathogenesis is complicated and poorly understood. Immune system abnormalities play a key role in the pathophysiology of bronchial asthma [54]. Classical immunology considers Th_1_/Th_2_ imbalance as a key factor in the pathogenesis of asthma and airway inflammation in general [55], which can promote eosinophil aggregation through the release of inflammatory cytokines that stimulate IgE synthesis and inflammatory cell infiltration [56]. IgE is an essential class of immunoglobulins in the human body, which mediates type I hypersensitivity reactions, and allergens that bind to IgE antibodies cause mast cells to degranulate, releasing cytokines, which causes illness [57]. As a result, high serum IgE concentrations are directly related to disease severity [58,59]. ICAM-1 has long been known to mediate cell-to-cell contacts in antigen presentation and intra- and extracellular signaling pathways, and it has been linked to asthma [60]. To investigate the effect of Cur-RH60/F127-MMs on airway inflammation in a mouse asthma model, researchers used OVA induction. In this study, we discovered that Cur-RH60/F127-MMs reduced the number of monocytes, lymphocytes, and granulocytes in the whole blood of OVA-induced asthmatic mice, as well as the release of inflammatory cytokines IL-6 and IL-10 and the inflammatory mediator NO in BALF. HE sections of lung tissue also verified that Cur-RH60/F 127-MMs reduced tissue inflammatory exudation, indicating that Cur-RH60/F127-MMs reduced the inflammation induced by bronchial asthma in OVA-sensitized mice. Furthermore, Cur-RH60/F127-MMs also dramatically reduced IgE and TNF-α levels in BALF of OVA mice, showing that the Th_1_/Th_2_ immune system imbalance in mice caused by OVA-sensitized bronchial asthma was improved.

Studies have also recently revealed that asthma is related to the biological mechanism of oxidation [61]. As a natural polyphenol compound, Curcumin has shown remarkable antioxidant potential [28,62]. 1,1-Diphenyl-2-bitter hydrazine (DPPH) is a stable nitrogen-centered radical, and the scavenging activity of this radical can be used to evaluate the antioxidant activity of drugs. In the anti-oxidation test, Cur-RH60/F127-MMs demonstrated higher antioxidant activity than Cur-API. According to recent research, peroxides and their consequent lipid peroxidation reactions are implicated in the inflammatory response process in asthmatic lung tissue and worsen lung histological alterations [63]. This study showed that Cur-RH60/F127-MMs dramatically reduced MDA content and boosted SOD activity in mouse lung tissues. The lipid peroxidation reaction was reduced, which protected lung tissue from inflammatory damage induced by the asthma airway.

## 5. Conclusions

In our study, we successfully prepared Cur-RH60/F127-MMs with Cur loaded in their cavity. MMs are a good candidate for drug loading because of their safety, stability, and good biocompatibility, especially for those drugs with low solubility. In this study, we greatly improved the solubility, bioavailability, and biological activities of Cur. The present study suggested that the treatment of Cur-Cr RH60/F127-MMs had a good regulating effect on airway inflammation. Therefore, Cur-RH60/F127-MMs might be used as a promising new candidate for the delivery of Cur to treat asthma.

## Figures and Tables

**Figure 1 pharmaceutics-15-02710-f001:**
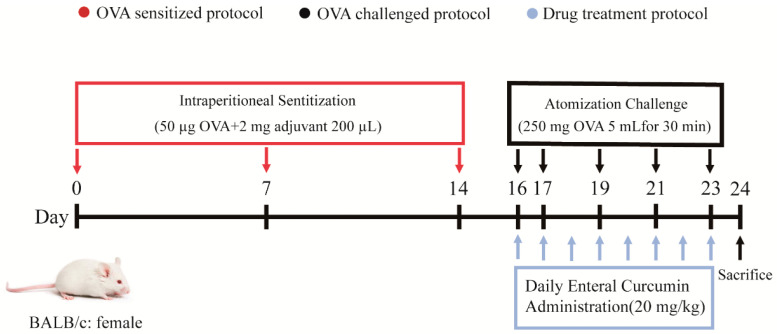
Schematic diagram of OVA-induced asthma model in BALB/c mice.

**Figure 2 pharmaceutics-15-02710-f002:**
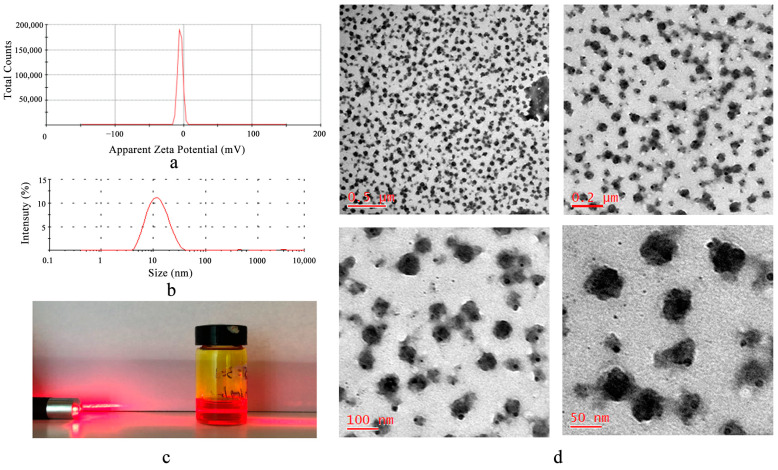
Zeta potential (**a**); particle size (**b**); Tyndall effect image (**c**); and TEM micrographs (**d**) of Cur-RH60/F127-MMs.

**Figure 3 pharmaceutics-15-02710-f003:**
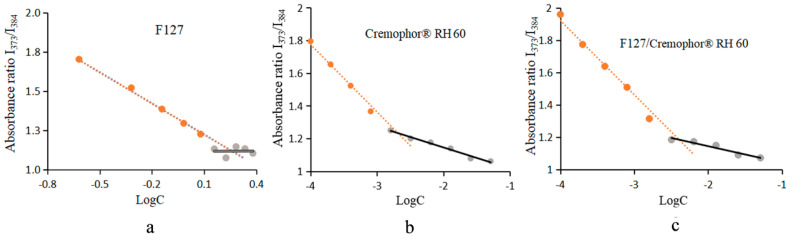
CMC values correspond to F 127 (**a**), RH60 (**b**), and RH60-MMs at I_373nm_/I_384nm_ (**c**) (*n* = 3).

**Figure 4 pharmaceutics-15-02710-f004:**
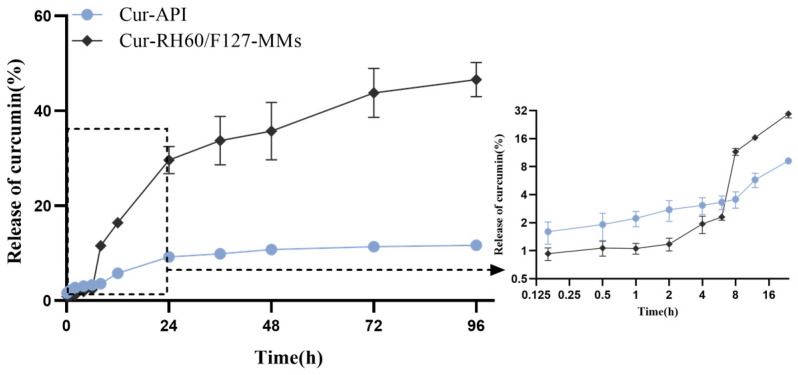
The release profiles of Cur-API and Cur-RH60/F127-MMs (the inset graph in the right is a detail of the main graph, mean ± SD, *n* = 6).

**Figure 5 pharmaceutics-15-02710-f005:**
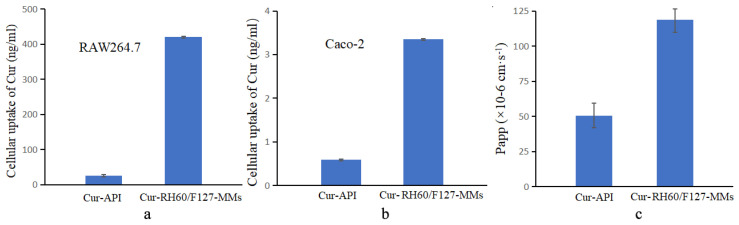
Comparative study on intracellular uptake of RAW 264.7 cells (**a**), Caco-2 cells (**b**) and transportation in Caco-2 cells (**c**) of Cur-API and Cur-RH60/F127-MMs (mean ± SD, *n* = 3).

**Figure 6 pharmaceutics-15-02710-f006:**
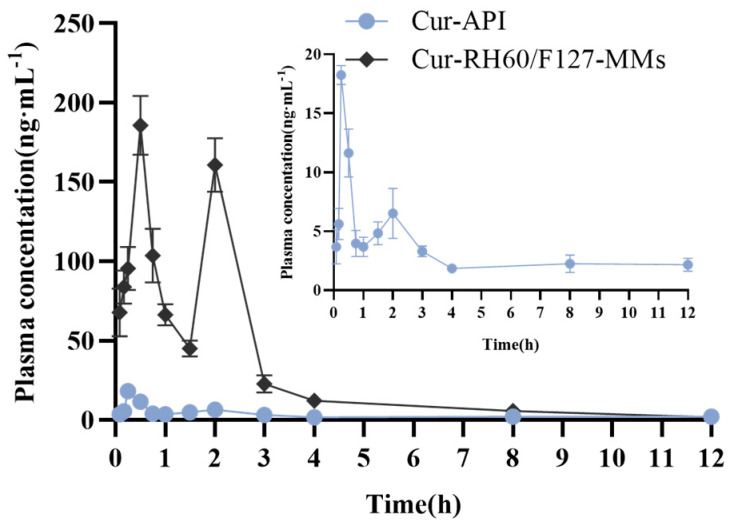
Mean plasma concentration–time curves after intragastric administration of Cur-API and Cur-RH60/F127-MMs of 60 mg/Kg in rats (the inset graph in the right is a detail of the main graph, mean ± SD, *n* = 6).

**Figure 7 pharmaceutics-15-02710-f007:**
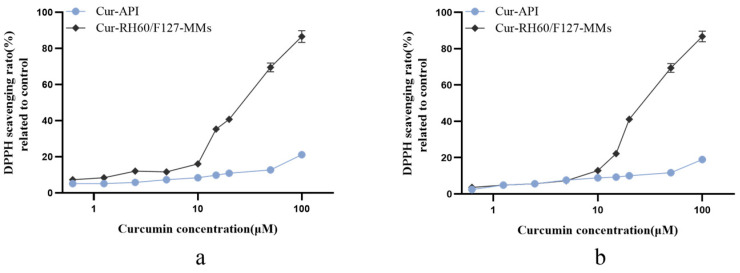
Effect of DPPH scavenging ability of Cur-API and Cur-RH60/F127-MMs (**a**), and that after 10 h of irradiation under UV light (30 W) (**b**) (mean ± SD, *n* = 6).

**Figure 8 pharmaceutics-15-02710-f008:**
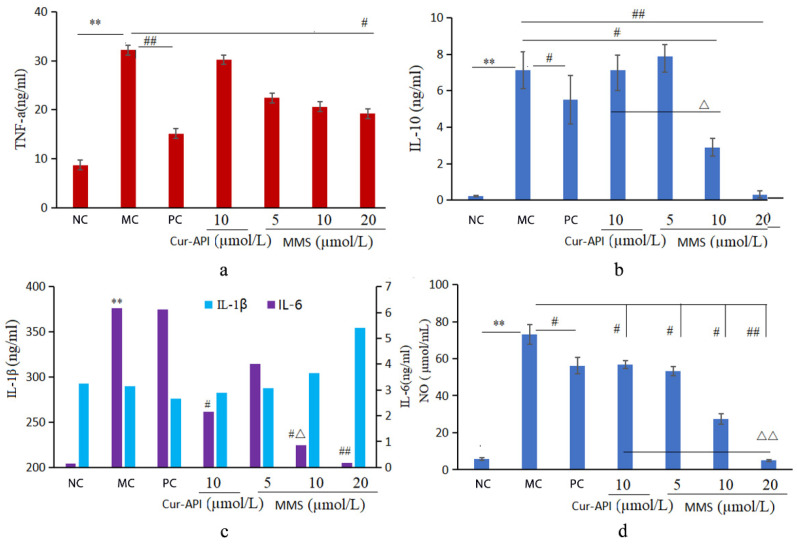
Effect of Cur-API and Cur-RH60/F127-MMs on TNF-α (**a**), IL-10 (**b**), IL-6 and IL-1β (**c**), and NO (**d**) levels of RAW264.7 cells (mean ± SD, *n* = 3). ** *p <* 0.01, compared with normal control group; ^#^
*p <* 0.05, ^##^
*p <* 0.01, compared with model control group; ^△^
*p <* 0.05, ^△△^
*p <* 0.01.

**Figure 9 pharmaceutics-15-02710-f009:**
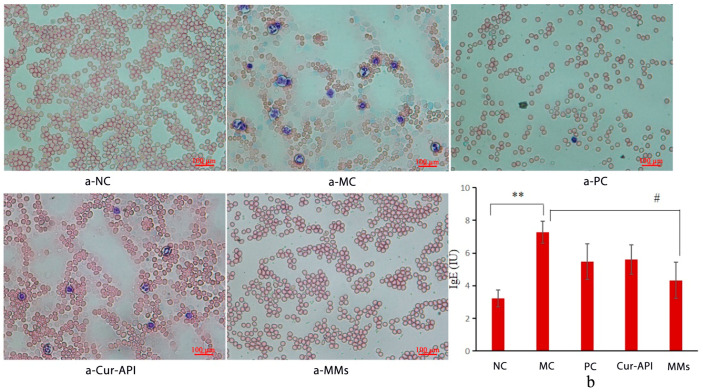
Effects of Cur-API and Cur-RH60/F127-MMs on whole blood cell counts ((**a**): a-NC; a-MC; a-PC; a-Cur-API; a-Cur-RH60/F127-MMs) and the level of IgE (**b**) (mean ± SD, *n* = 10). ** *p <* 0.01, compared with normal Control group; ^#^
*p <* 0.05 compared with model control group.

**Figure 10 pharmaceutics-15-02710-f010:**
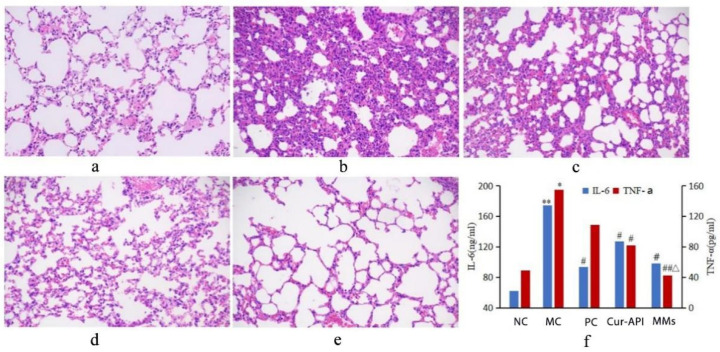
HE staining of lung tissue (×200, (**a**): NC; (**b**): MC; (**c**): PC; (**d**): Cur-API; (**e**): Cur-RH60/F127-MMs) and the levels of IL-6 and TNF-α in BALF (**f**) (mean ± SD, *n* = 10). * *p <* 0.05, ** *p <* 0.01, compared with normal control group; ^#^
*p <* 0.05, ^##^
*p <* 0.01, compared with model control group; ^△^
*p <* 0.05, compared with Cur-API group.

**Figure 11 pharmaceutics-15-02710-f011:**
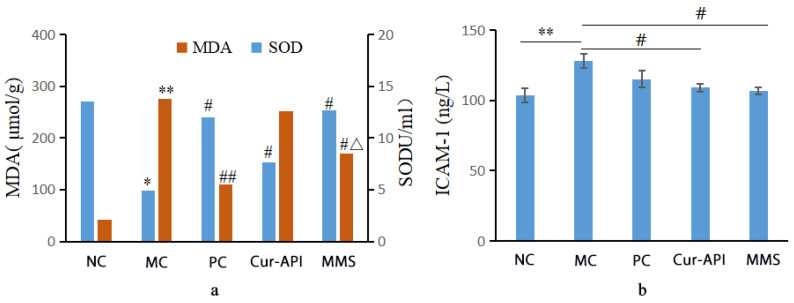
Effect of SOD activity, MDA concentration (**a**), and intercellular adhesion factor ICAM-1 (**b**) in the lung tissue of the groups (mean ± SD, *n* = 10). * *p* < 0.05, ** *p* < 0.01, compared with normal control group; ^#^ *p* < 0.05, ^##^ *p* < 0.01, compared with model control group; ^△^ *p* < 0.05, compared with Cur-API group.

**Table 1 pharmaceutics-15-02710-t001:** Mathematical models of Cur-API and Cur-RH60/F127-MMs to dissolution profiles (*n* = 6).

Mathematical Model		Equation	Rsqr_adj
Cur-API	Zero-order	F = 0.1163t + 3.1681	0.8103
	First-order	ln(100 − F) = −0.0013t + 4.5729	0.8163
	Higuchi	F = 1.2523t^1/2^ + 1.0855	0.9383
	Ritger–Peppas	F = 2.1951t^0.3881^	0.9424
	Hixson–Crowell	(100 − F)^1/3^ = −0.0019t + 4.592	0.8143
Cur-RH60/F127-MMs	Zero-order	F = 0.5627t + 4.6594	0.8558
	First-order	ln(100 − F) = −0.0076t + 4.5603	0.8969
	Higuchi	F = 5.9356t^1/2^ − 4.936	0.9506
	Ritger–Peppas	F = 3.4830t^0.5993^	0.9319
	Hixson–Crowell	(100 − F)^1/3^ = −0.0106t + 0.8835	0.8143

**Table 2 pharmaceutics-15-02710-t002:** Main pharmacokinetic parameters of Cur after intragastric administration of Cur-API and Cur-RH60/F127-MMs of 60 mg/Kg in rats (mean ± SD, *n* = 6).

Parameters	Cur-API	Cur-RH60/F127-MMs
AUC_0–12_/(ng·h·mL^−1^)	37.7 ± 17.22	348.32 ± 101.42 **
AUC_0–∞_/(ng·h·mL^−1^)	54.21 ± 28.6	353.64 ± 106.33 *
T_max_/h	0.31 ± 0.13	0.88 ± 0.75 *
C_max_/(ng·mL^−1^)	18.50 ± 1.81	199.84 ± 39.05 **
T_1/2_/h	5.16 ± 2.76	1.98 ± 0.91
MRT_(0–12)_/h	4.37 ± 0.88	2.11 ± 0.4 **
CLz/F (L·h^−1^·kg^−1^)	982.85 ± 314.28	181.97 ± 54.58
Vz/F (L·kg^−1^)	5235.15 ± 1049.01	495.40 ± 194.53

* *p* < 0.05, ** *p* < 0.01, compared with Cur-API.

**Table 3 pharmaceutics-15-02710-t003:** Effect of Cur-API and Cur-RH60/F127-MMs on whole blood cell counts (mean ± SD, *n* = 10 × 10^9^/L).

Group	Total Cells	Monocytes	Lymphocytes	Granulocytes
Normal Control	5.43 ± 0.31	0.13 ± 0.05	4.13 ± 0.17	1.17 ± 0.12
Model Control	35.03 ± 6.79 **	1.8 ± 0.45 **	18.93 ± 4.23 **	14.3 ± 2.61 **
Positive Control (1.4 g·kg^−1^)	11.03 ± 0.25 ^##^	0.67 ± 0.09 ^##^	4.1 ± 0.50 ^##^	6.27 ± 0.58 ^##^
Cur-API (20 mg·kg^−1^)	9.46 ± 1.44 ^##^	0.26 ± 0.07 ^##^	7.04 ± 1.30 ^##^	2.16 ± 0.37 ^##^
MMs (20 mg·kg^−1^)	7.6 ± 2.57 ^##^	0.18 ± 0.04 ^##^	5.15 ± 2.33 ^##^	2.28 ± 0.47 ^##^

** *p* < 0.01, compared with normal Control group; ^##^ *p* < 0.01, compared with model control group.

## Data Availability

The authors confirm that the data supporting the findings of this study are available within the article.

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
