# Peer review of "Curcumin-Loaded RH60/F127 Mixed Micelles: Characterization, Biopharmaceutical Characters and Anti-Inflammatory Modulation of Airway Inflammation"

_pharmaceutics, 2023, doi:10.3390/pharmaceutics15122710_

Round 1

Reviewer 1 Report

Comments and Suggestions for Authors

 Xinli Wang and co-authors have submitted manuscript, wherein encapsulation of curcumin has been performed in order to enhance the solubilitya nd biovaialbility. In this regards,  RH60/F127 based polymeric micelles have been designed and evaluated for various physico-chemical and  in-vitro characeterization. Further, asthma model employed to check its biological efficacy. Taken together, the manuscript is well-organised and covers all crucial aspects. Therefore, it can be accepted for the publication.

Good Luck!

Comments on the Quality of English Language

English throughout the manuscript was fine. 

Author Response

Thank you for your comments concerning our manuscript entitled “Curcumin Loaded RH60/F127 Mixed Polymeric Micelles: Characterization, Biopharmaceutical Characters and Anti-inflammatory Modulation of Airway Inflammation” (ID: pharmaceutics-2631104). Those comments are all valuable, as well as the important guiding significance to our researches. The manuscript has been checked using the paid editing service offered by Editorial Office. We hope the revised manuscript could be acceptable for you.

Reviewer 2 Report

Comments and Suggestions for Authors

The authors investigated curcumin-loaded polymeric mixed micelles (cur-MM) for oral delivery. These cur-MM were characterized with several physical-chemistry methods including transmission electron microscopy (TEM). The results on the biological effect showed that Cur-MMs prominently inhibited the production of proinflammatory cytokines. In addition, Cur- -MMs enormously reduced airway hyper responsiveness and inflammation in vivo experimental asthma model.

I have the following minor suggestions for improvement of the article:

1.       Abstract: it could be rewritten in a smoother way to avoid declaration of Objective, Methods, and Results. The abbreviations such as Cur and RH60 can be explicitly introduced.

2.       Fig.2b: the x-axis scale should be in [nm] not in (dm).

3.       Table 1 is not well formatted. The cells for the different items are merged and the separator line is not visible. To address the problem, please provide the content of the table, including the headers and the data for each item in a clear way.

4.       The authors may consider expanding the discussion with a recent paper on curcumin multidrug nanoparticles for oral delivery that is closely related to the investigated topic : “Curcumin- and Fish Oil-Loaded Spongosome and Cubosome Nanoparticles with Neuroprotective Potential against H2O2-Induced Oxidative Stress in Differentiated Human SH-SY5Y Cells”, ACS Omega 2019, 4, 2, 3061–3073

Comments on the Quality of English Language

English needs careful proofreading by a professional English speaking writer.

Reviewer 3 Report

Comments and Suggestions for Authors

I have read the manuscript “Curcumin Loaded RH60/F127 Mixed Polymeric Micelles: Characterization, Biopharmaceutical Characters and Anti-inflammatory Modulation of Airway Inflammation” by Xinli Wang et al. (MS # pharmaceutics-2631104) submitted for the publication in Pharmaceutics.

In their manuscript the authors reported the results of their work on the preparation and characterization of polymer micelles loaded with Curcumin. They also investigated the effects of CUR on a mice model of bronchial asthma.

The topic of the manuscript is certainly interesting for the potential applications, but it needs some major revisions before its publication in Pharmaceutics. In particular:

1.      English needs some revisions. There some misprints (e.g. caster oil, cremophor and pluronic need capital initials, originc8.0, flower, line 359, line 372,…), some sentences sound fragments (e.g. lines 19-20, 216, 295, … ), and section 2.6 sounds as an instrument manual.

2.      The MMs were found to have a zeta potential of -7.62 mV, which is a rather low value. What about the long-term stability of MMs? How does the average size change with time?

3.      Line 298 and 473: at higher magnifications of TEM pictures, nanoparticles look like as aggregates of smaller primary spherical nanoparticles in agreement with DLS measurements.

4.      Figures 4 and 6: please, describe the insets.

5.      Table 3: please, define blood components.

6.      Discussion and Table 2: a more detailed discussion for CUR release should be given on the basis of the found values for different parameters present in the fitting equations.

Comments on the Quality of English Language

English needs some revisions. There some misprints (e.g. caster oil, cremophor and pluronic need capital initials, originc8.0, flower, line 359, line 372,…), some sentences sound fragments (e.g. lines 19-20, 216, 295, … ), and section 2.6 sounds as an instrument manual.

Reviewer 4 Report

Comments and Suggestions for Authors

The article by Wang, X. et al. about the generation, description and effects in asthma condition of micelles for curcumin appears scientifically correct and well organized to my eyes.

I have some major comment to make:

- You refer to nanomicelles (abstract, line 19), whose acronym should be “NM” or to micromicelles (MM)?? This is not clarified in the text. The new text I suggest immediately below should explain this point.

- The introduction is written as a concatenation of various definitions, and the end paragraph is a description of the experiment. I miss a lot of text regarding the actual INTRODUCTION, you should describe why you are making this experiment and its relevance, and the usefulness of this study. For example, the first paragraph in the discussion might be in the introduction for this purpose (with some adaptation, of course).

- I think the histopathological description should be greatly broadened. You currently have convincing figures, but they are poorly described. I include some suggestions as minor comments.

There are also some minor comments:

- Abstract. Line 27. Check the % symbol. Also in line 116. May be other examples. The same for degrees. These symbols are situated immediately after the number, not after the parentheses.

- Animals and cell culture. Lines 111, 113. Please include the number of animals before their name, not after it.

- Determination of Blank MMs. Lines 162-163. Please, use past tense.

- Pharmacokinetic study. You already mentioned these rats in point 2.2.

- Histopathology of the lungs, line 268. You already mentioned “the trachea” as the subject one time in the sentence, remove the other.

- Materials and methods (general). In point 2.2 you mention that you purchased 100 BALB mice, but in point 2.12 it seems you only used 50 mice. I am a bit confused, I didn´t found the role of the other 50 mice.

- Results, line 336. I feel something is missing before “Cur-RH60/F127-MM are released by slow diffusion”.

- Results, figure 4. You should explain in the text legend that the small graph in the right is a detail of the main graph.

- Results. Table 1 is poorly explained.

- Results, line 345. You shouldn´t begin a sentence with “And”. The same in line 399.

- Results, line 384. This paragraph, linked to figure 8, seems a separate subsection… would be point 3.7 and, in this case, following subsections should be renumbered.

- Results, lines 403-405. This sentence should be in discussion, as it is not a result.

- Results, table 3. You should explain the abbreviations in the legend.

- Results. Figure 9 is poorly described. You should explain in the legend the meaning of the headings under the different images. In addition, you should explain what is depicted in each image, this is a pharmaceutical journal. I mean, in the first figure there are only erythrocytes, in the second there are several neutrophils and some lymphocytes, in the third very few cells… You should explain all of this in a new, broader, legend. The same can be applied to figure 10 (although in this case a brief description is already made in the main text, this is insufficient and you do not try to describe each image).

- Results. I think a different subsection should be dedicated to the lung histopathological study. It seems strange the beginning of this in line 416.

- Results, line 439. ICAM-1 is wrongly spelled.

- Results (general). Although the figures and tables seem relevant and well placed, you should mention each figure in the main text. For example Figures 6 or 7 are not mentioned in the main text.

- Discussion. Line 492. Please, remove the first “and” of the sentence. However, for lines 499-501, you should remove the second and (“cytokines, causing illness”).  You can remove “While” in line 509.

Comments on the Quality of English Language

The language is generally fine, with various minor issues. I mention some of them. A language review is necessary.   

Round 2

Reviewer 3 Report

Comments and Suggestions for Authors

The authors have answered in a satisfactory manner to all comments: Th emanuscript can be published as it is

Author Response

Thanks a lot for your  questions last time, and I have provided the revised manuscript again.

Reviewer 4 Report

Comments and Suggestions for Authors

The article by Wang, X. et al. about the generation, description and effects in asthma condition of micelles for curcumin was improved according to reviewers suggestions, including mines. I would like to mention that, although I didn´t mentioned it specifically in my review, the abstract looks way more attractive this way (actually, when reading the first version I thought that it was not particularly attractive). However, I think my comments related with the histological study were ignored, and that´s the reason for which I keep my previous evaluation asking for major revision of the manuscript.

I have some major comment to make:

- Although I still miss a lot of text (at least a whole paragraph) in the introduction, you should in any case first describe the curcumin and then, the Cremophor. To my eyes, the Cremophor is just the tool to have the curcumin delivered, and as such, the curcumin is the first thing to explain.

- I think the histopathological description should (still) be greatly broadened. I give more details in several “minor comments”.

There are also some minor comments:

- Animals and cell culture. Lines 136, 138. Please include the number of animals before their name (12 male Sprague-Dawley rats, for example), in the revised version you just removed the number.

- Determination of Blank MMs. Lines 188. Then, the solution WAS ADDED.

- Pharmacokinetic study. Line 232. If you mention the number of rats in point 2.2, then you just need to write “the SD rats were…” here. The same for the mice in section 2.12.

- 2.11. Effect of Cur-RH60/F127-MMs. Line 262. What´s the difference between “blank control group” and “model group”? I understand blank control, but I am unsure about that´s model group. You already define the positive control group.

- 2.12. Effect of Cur-RH60/F127-MMs on airway inflammation. Line 275. Here you define positive control group as 1,4 g/kg of Keke tablets, in point 1.11 as 2microg of dexamethasone DEX. I guess you refer to the same thing. I think the definition in point 1.11 is better and should be in all the text. You probably should define Keke tablets as dexamethasone in line 117 of materials and methods.

- Histopathology of the lungs, line 299. I think the correct form to refer the dissection is passive (“the lobe was taken and HE staining was performed); I don´t like “we took”.

- Results, section 3.8. Line 479. I think you refer to figure 9b. In addition, the separation of the different images in Figure 9 is still confusing. You present here a figure with 6 images, but you pretend to have only figures 9a and 9b. You should include, in any case, an insert in each image to name them from 9a to 9f. When you refer to them in the text you can include “figures 9a-e” and “9f” if you want. Please, include in the legend that NC means normal control. In addition, you should briefly explain what cells are depicted in each image in the text legend. I will just copy the suggestion made in my previous report: “in the first figure there are only erythrocytes, in the second there are several neutrophils and some lymphocytes, in the third very few cells…” You should explain all of this in a new, broader, legend.

- Results, section 3.9. Lines 494-497. This sentence is very long and has a poor grammar (while something has wathever, other thing has other characteristic, in the text you only mention lung tissue in the model groups).You should describe all the groups, they are still poorly described. Moreover, the same comments in section 3.8 can be applied to figure 10 (although in this case a brief description is already made in the main text, this is insufficient and you do not try to describe each image).

Comments on the Quality of English Language

I see very few language issues.

Author Response

Dear reviwers and editors,

Thank you for your comments concerning our manuscript entitled “Curcumin Loaded RH60/F127 Mixed Polymeric Micelles: Characterization, Biopharmaceutical Characters and Anti-inflammatory Modulation of Airway Inflammation” (ID: pharmaceutics-2631104). We are sorry that the previous revision and reply were not accepted for you. We have studied comments carefully and have made corrections which we hope meet with approval again. Revised portion are marked in red and blue in the paper. We hope the new revised manuscript could be acceptable for you. The main corrections in the paper and the responses to the reviewer’s comments are as follows:

- I think the histopathological description should (still) be greatly broadened. I give more details in several “minor comments”.

Response: We sincerely appreciate the valuable comments. The histopathological description bas been broadened in the revised manuscript.

There are also some minor comments:

- Animals and cell culture. Lines 136, 138. Please include the number of animals before their name (12 male Sprague-Dawley rats, for example), in the revised version you just removed the number.

Response: Thanks for your suggestion. We are sorry for our misunderstanding. We have revised in the revised manuscript.

- Determination of Blank MMs. Lines 188. Then, the solution WAS ADDED.

Response: We are sorry for our mistake. We have revised in the revised manuscript.

- Pharmacokinetic study. Line 232. If you mention the number of rats in point 2.2, then you just need to write “the SD rats were…” here. The same for the mice in section 2.12.

Response: Thanks for your suggestion. We have deleted the duplicate parts in the revised manuscript.

- 2.11. Effect of Cur-RH60/F127-MMs. Line 262. What´s the difference between “blank control group” and “model group”? I understand blank control, but I am unsure about that´s model group. You already define the positive control group.

Response: Thanks for your careful checks, we have redefine nomal control group(NC), model control group(MC), positive control group (PC). Model control group(MC) is given LPS (1 g/mL) and INF-γ without any treatment.

- 2.12. Effect of Cur-RH60/F127-MMs on airway inflammation. Line 275. Here you define positive control group as 1,4 g/kg of Keke tablets, in point 1.11 as 2microg of dexamethasone DEX. I guess you refer to the same thing. I think the definition in point 1.11 is better and should be in all the text. You probably should define Keke tablets as dexamethasone in line 117 of materials and methods.

Response: Thanks for your careful checks. Keke Tablet is an over-the-counter Chinese medicine commonly used to treat asthma in China. So we chose it as a positive drug but not dexamethasone.

- Histopathology of the lungs, line 299. I think the correct form to refer the dissection is passive (“the lobe was taken and HE staining was performed); I don´t like “we took”.

Response: Thanks for your valuable suggestion. We have removed this sentence to the discussion part in the revised manuscript.

- Results, section 3.8. Line 479. I think you refer to figure 9b. In addition, the separation of the different images in Figure 9 is still confusing. You present here a figure with 6 images, but you pretend to have only figures 9a and 9b. You should include, in any case, an insert in each image to name them from 9a to 9f. When you refer to them in the text you can include “figures 9a-e” and “9f” if you want. Please, include in the legend that NC means normal control. In addition, you should briefly explain what cells are depicted in each image in the text legend. I will just copy the suggestion made in my previous report: “in the first figure there are only erythrocytes, in the second there are several neutrophils and some lymphocytes, in the third very few cells…” You should explain all of this in a new, broader, legend.

Response: We sincerely appreciate the valuable comments. We have included an insert in each image in Figure 9 and named them 9a to 9f, and explained what cells are depicted in each image in the text legend in the revised manuscript.

Results, section 3.9. Lines 494-497. This sentence is very long and has a poor grammar (while something has wathever, other thing has other characteristic, in the text you only mention lung tissue in the model groups).You should describe all the groups, they are still poorly described. Moreover, the same comments in section 3.8 can be applied to figure 10 (although in this case a brief description is already made in the main text, this is insufficient and you do not try to describe each image).

Response: We sincerely appreciate the valuable comments. Section 3.9 has been re-edited in the revised manuscript.

Yours sincerely,
Xinli Wang

E-mail: [email protected]
Corresponding author:
Name: Xin-li Liang
E-mail: [email protected]
